# miR-24-3p Is Antiviral Against SARS-CoV-2 by Downregulating Critical Host Entry Factors

**DOI:** 10.3390/v16121844

**Published:** 2024-11-28

**Authors:** Parrish Evers, Spencer M. Uguccioni, Nadine Ahmed, Magen E. Francis, Alyson A. Kelvin, John P. Pezacki

**Affiliations:** 1Department of Chemistry and Biomolecular Sciences, University of Ottawa, Ottawa, ON K1N 6N6, Canada; pever068@uottawa.ca (P.E.); suguc084@uottawa.ca (S.M.U.);; 2Vaccine and Infectious Disease Organization (VIDO), University of Saskatchewan, Saskatoon, SK S7N 5E3, Canada; m.francis@usask.ca (M.E.F.); alyson.kelvin@usask.ca (A.A.K.); 3Department of Biochemistry, Microbiology, and Immunology, University of Saskatchewan, Saskatoon, SK S7N 5E3, Canada

**Keywords:** SARS-CoV-2, COVID-19, microRNA, miR-24, miR-24-3p, HCoV-229E, NRP1, NRP2, furin, SREBP2

## Abstract

Despite all the progress in treating SARS-CoV-2, escape mutants to current therapies remain a constant concern. Promising alternative treatments for current and future coronaviruses are those that limit escape mutants by inhibiting multiple pathogenic targets, analogous to the current strategies for treating HCV and HIV. With increasing popularity and ease of manufacturing of RNA technologies for vaccines and drugs, therapeutic microRNAs represent a promising option. In the present work, miR-24-3p was identified to inhibit SARS-CoV-2 entry, replication, and production; furthermore, this inhibition was retained against common mutations improving SARS-CoV-2 fitness. To determine the mechanism of action, bioinformatic tools were employed, identifying numerous potential effectors promoting infection targeted by miR-24-3p. Of these targets, several key host proteins for priming and facilitating SARS-CoV-2 entry were identified: furin, NRP1, NRP2, and SREBP2. With further experimental analysis, we show that miR-24-3p directly downregulates these viral entry factors to impede infection when producing virions and when infecting the target cell. Furthermore, we compare the findings with coronavirus, HCoV-229E, which relies on different factors strengthening the miR-24-3p mechanism. Taken together, the following work suggests that miR-24-3p could be an avenue to treat current coronaviruses and those likely to emerge.

## 1. Introduction

Nearing five years since the beginning of the COVID-19 pandemic, a long-term shift to endemic SARS-CoV-2 is expected and is beginning to be modeled using data from other well-known respiratory illnesses [1]. As SARS-CoV-2 incidence persists, it is important to consider how we can continue to improve upon current antiviral therapies and search for antiviral tools that act more broadly on multiple members of the coronaviridae family. The latter seems increasingly relevant considering COVID-19 is the third major coronavirus outbreak in the last 20 years [2], and by far the costliest; this is evidenced by the >700 million confirmed cases [3] and the multi-trillion-dollar economic impact [4]. Moreover, the appearance of escape mutant variants against vaccines [5,6] and antivirals [7] remains an ongoing concern with treating SARS-CoV-2. Therefore, it is critical to explore different strategies for combating coronaviruses that are currently circulating and those that are likely to emerge in the future.

MicroRNAs (miRNAs) are small non-coding RNAs (ncRNAs) that associate with the RNA-induced silencing complex (RISC) to post-transcriptionally regulate genes [8]. Specific miRNAs may serve critical or more nuanced roles [9] depending on factors such as miRNA-mRNA complementarity and the expression levels of the miRNA or its target mRNAs [10]. Unsurprisingly, there are numerous examples of miRNAs being differentially regulated during viral infections [11,12,13,14,15], where they may have mechanisms promoting or inhibiting infection [11,15,16,17]; furthermore, some viruses, like Epstein–Barr virus (EBV) or human immunodeficiency virus (HIV), even encode miRNAs in their viral genome [16]. Therefore, the use of miRNA and miRNA-targeting therapeutics appears to be an attractive antiviral tool and has previously been explored in the notable example of Miravirsen to treat hepatitis C virus (HCV) [18,19].

miR-24-3p is a commonly studied miRNA previously characterized for its role in regulating lipids [20] and its impact on the MAPK pathway [21]. Interestingly, there has been accumulating evidence for a relationship between miR-24-3p and infection of viruses; for instance, miR-24-3p overexpression has been shown to reduce HCV proliferation [22], reduce the spread of H5N1 influenza A [23], and decrease infectivity of vascular stomatitis virus (VSV) [24]. Conversely, miR-24-3p seems to promote the infection of herpes simplex virus 1 (HSV-1) [25] and respiratory syncytial virus (RSV) [26] due to decreased viral replication in response to inhibiting endogenous miR-24-3p. The findings with RSV appear to be mixed, however, with more recent findings that miR-24-3p overexpression disrupts RSV replication [27]. The current literature surrounding miR-24-3p and SARS-CoV-2, however, remains sparse aside from a report that extracellular vesicular miR-24-3p is decreased in COVID-19 patients [28].

Given the functional implications of miR-24-3p on viruses generally, we sought to elucidate if miR-24-3p has antiviral properties against SARS-CoV-2 and HCoV-229E. We determined that miR-24-3p was inhibitory toward SARS-CoV-2. To elucidate this mechanism, we then turned to the bioinformatic prediction tool miRDB [29], which identified several factors involved in viral infections. We then selected targets previously characterized to affect viral entry: furin, neuropilin-2 (NRP2), and sterol regulatory element binding protein 2 (SREBF2 or SREBP2) for further validation.

Furin is a well-studied proprotein protease, residing mainly in the golgi apparatus, and where it is responsible for cleaving a polybasic R-X-X-R (though R-X-R/K-R is preferred) motif within target proteins [30,31]. From an infectious disease standpoint, furin has previously been characterized in activating anthrax toxin [30], priming produced HIV virions for viral entry [32], cleaving hemagglutinin, an essential process for influenza infection [33], and facilitating the infection of MERS-CoV [34]. In the context of SARS-CoV-2, furin cleaves the R-R-A-R sequence joining the S1 and S2 regions of the S protein in cells producing new virions [31,35], and this has been suggested to offer an evolutionary advantage to SARS-CoV-2 [36,37]. Cleavage at the S1/S2 site is a necessary step before cleavage at the S2′ site by cell surface proteases (ex: TMPRSS2) or lysosomal proteases (ex: cathepsin L), enabling membrane fusion during viral entry [38].

NRP1 has been highly discussed since the beginning of the COVID-19 pandemic as an entry factor for SARS-CoV-2 [39,40,41], and along with NRP2, was identified as being upregulated in COVID-19 patient samples [42]. Moreover, in one of the earlier reports, both NRP1 and 2 were shown to bind the S1 subunit [39], and this similarity in binding is supported by previous reports of overlapping targets for NRP1 and NRP2 [43]; however, it is important to note that non-overlapping targets exist [43], likely due to the 47% amino acid sequence homology (NP_003864.5; NP_957718.1). Nonetheless, NRP2 is an entry receptor for human cytomegalovirus (HCMV) [44,45], an entry factor for Lujo virus [46] and has recently been identified to enhance SARS-CoV-2 infection when overexpressed [47]. Interestingly, miR-24-3p has been shown to reduce NRP1 using a 3′ UTR luciferase assay [48]; however, this work did not assess the experimental effect of miR-24-3p on SARS-CoV-2, nor were the implications of miR-24-3p on NRP2 explored. Taken together, these previous works merit a further investigation of NRP2 in the miR-24-3p-mediated attenuation of SARS-CoV-2.

SREBP2 has previously been characterized to promote hepatitis C virus (HCV) infection [49,50], and our group has shown that it is downregulated by miR-185 to attenuate HCV infection [14] and likely contributes to the miR-185-mediated repression of SARS-CoV-2 [17]. It has been suggested that SREBP2 mechanistically affects viruses through the remodeling of the lipidome (including membranes) and cholesterol biosynthesis [51]; accordingly, cholesterol has been identified as an important entry factor for SARS-CoV-2 while SREBP2 (along with SREBP1 and 1c) has been identified to promote viral infection in MERS-CoV [52]. Given this information, it is likely that SREBP2 downregulation is an aspect of miR-24-3p viral inhibition.

To study these identified host entry factors in isolated study from the additional steps of viral infection, we employed a SARS-CoV-2 spike (S) pseudovirus [35,53] previously used by our group [17]. We determined that miR-24-3p decreases SARS-CoV-2 pseudovirion entry and that the treatment is resistant to common S mutants. Furthermore, the miR-24-3p mediated attenuation of these factors resulted in reduced viral entry and resulted in less infectious pseudovirions. The findings of the present study suggest that miR-24-3p inhibits infection of SARS-CoV-2 by downregulating key entry factors and suggests that miR-24-3p could be a useful addition to current SARS-CoV-2 treatment regimens or future emerging coronaviruses.

## 2. Materials and Methods

### 2.1. Reagents and Cell Culture

Huh7 human hepatoma cell lines were kindly gifted by Dr. Charles M. Rice (Rockefeller University, New York, NY, USA). Huh7 were cultured and maintained in Dulbecco’s Modified Eagle Medium (DMEM; Invitrogen, Waltham, MA, USA) supplemented with 10% fetal bovine serum (FBS; Wisent Bio Products, Saint-Jean-Baptiste, QC, Canada). HEK293T cells (CRL-3216), A549 (CCL-185) and Calu-3 cells (HTB-55) were purchased from ATCC and were cultured in DMEM supplemented with 10% FBS. A549 cells stably expressing ACE2 (A549-ACE2) cells were generated using a lentiviral, pLENTI_hACE2_PURO plasmid (Addgene #155295), a gift from Prof. Raffaele De Francesco. Puromycin was used for selection and stable expression was confirmed via western blotting. miR-24-3p mimics, as well as negative control mimics (con-miR) were purchased from mirVana (Ambion, Austin, TX, USA). pCMV14-3X-Flag-SARS-CoV-2 S was a gift from Zhaohui Qian (Addgene plasmid #145780; http://n2t.net/addgene:145780 (accessed on 2 August 2022, RRID:Addgene_145780); RRID: Addgene 145780). The HIV-1 NL4-3 ΔEnv Vpr Luciferase Reporter Vector (pNL4-3.Luc.R-E-) was a kind gift from Dr. Benoit Barbeau from the University of Quebec in Montreal. SARS-CoV-2 isolate/Canada/ON/VIDO-01-2020, belonging to the original Wuhan strain, was used for SARS-CoV-2 infections. This virus was isolated from a patient at a Toronto hospital who had returned from Wuhan, China. The second passage viral stock was sequenced (GISAID—EPI_ISL_425177) to confirm the stability of the virus after culture in vDMEM on Vero-76 cells. All work with infectious SARS-CoV-2 virus was performed in a Containment Level 3 (CL3) facility at the Vaccine and Infectious Disease Organization (VIDO) (Saskatoon, SK, Canada). HCoV-229E was a gift from the laboratory of Dr. Maxime Berezovski (University of Ottawa, Ottawa, ON, Canada) (ATCC; VR-740). All work with HCoV-229E was performed in a Containment Level 2 (CL2) facility at the University of Ottawa.

### 2.2. Introduction of D614G and N501Y Mutations in SARS-CoV-2 Spike

Point mutations were introduced into the pCMV14-3X-Flag-SARS-CoV-2 S to generate the D614G and N501Y mutants using a Quick-change Lightning kit (Agilent, Santa Clara, CA, USA) as per the manufacturer’s protocol. The D614G mutation was introduced with the following primers: Forward: 5′ GTGCTGTACCAAGGCGTGAATTGCACAG-3′ Reverse: 5′-CTGTGCAATTCACGCCTTGGTACAGCAC-3′. The N501Y mutation was introduced with the primers: Forward: 5′-GGATTCCAGCCAACCTACGGCGTGGGTTACCAAC-3′ Reverse: 5′ GTTGGTAACCCACGCCGTAGGTTGGCTGGAATCC-3′. Mutations were confirmed by Sanger sequencing at Génome Québec.

### 2.3. Production of SARS-CoV-2 S Pseudovirus

Pseudovirus (pseudotyped viral particles) was generated according to previously published methods used by our lab and others [17,35,53]. In brief, HEK293T cells were cultured in 10 cm dishes, and were co-transfected with 2 µg of pcDNA3.1 plasmid containing codon-optimized cDNA for the SARS-CoV-2 S glycoprotein (or empty pcDNA3.1), and 4 µg of HIV-1-NL4-3 ∆Env Vpr Luciferase Reporter Vector (pNL4-3.Luc.R-E). A ratio of 1 µg plasmid to 1.2 µL Lipofectamine 2000 ensured efficient transfection of plasmids. 72 h post-transfection of these plasmids, supernatants containing pseudovirions were collected, centrifuged at 800× *g* for 5 min to remove cell debris, and then passed through a 0.45 µm filter.

### 2.4. miRNA Transfections in Pseudovirus-Producing Cells

To study the effect of a miRNA in the production of pseudovirions, HEK29T cells were cultured in 10 cm dishes, and were transfected with 50 nM of miRNA-24-3p mimic, or a negative control, con-miR mimic. A ratio of 1 µL 100 µM miRNA mimic to 2.5 µL Lipofectamine RNAiMax was used to ensure efficient transfection of miRNAs. Then, 24 h later, they were co-transfected with 2 µg of pcDNA3.1 plasmid containing codon-optimized cDNA for the SARS-CoV-2 S glycoprotein, and 4 µg of HIV-1-NL4-3 ∆Env Vpr Luciferase Reporter Vector (pNL4-3.Luc.R-E). Then, 72 h after the transfection of these plasmids, pseudovirions were isolated as described above. Cells that produced the pseudovirions were washed with PBS before being lysed by sonication in RIPA buffer (10 mM Tris-HCl, pH 8.0, 1 mM EDTA, 1% NP-40, 0.1% Sodium Deoxycholate, 0.1% SDS).

### 2.5. miRNA Transfections of Host Cells and Pseudovirus Entry Assays

To study the effect of a miRNA on the entry of pseudovirions, Huh7 cells and A549-ACE2 cells were seeded in 24-well plates and transfected the following day with 100 nM of miR-24-3p mimic, or a negative control miRNA mimic. Alternatively, Calu-3 cells were reverse transfected in 24-well plates with 100 nM of miR-24-3p mimic or a negative control, control miRNA mimic. Then, 24 h following the transfection of the miRNA, cells were transduced with 100 µL of supernatant containing pseudovirions. Spinfection at 800× *g* for 1 h was performed to ensure efficient viral attachment. For all cell lines, 48 h post-infection, cells were lysed in 1X Passive Lysis Buffer (Promega, Madison, WI, USA), and Luciferase activity was measured using a microplate reader. Luciferase signal was normalized against total protein concentration as determined by Bradford assay. All experiments were read in technical triplicates for at least three biological replicates.

### 2.6. Detection of Proteins by Western Blotting

Huh7 cells and HEK293T cells were lysed by sonication in RIPA buffer (10 mM Tris-HCl, pH 8.0, 1 mM EDTA, 1% NP-40, 0.1% Sodium Deoxycholate, 0.1% SDS). Following protein quantification by detergent-compatible (DC) assay (Bio-Rad, Hercules, CA, USA), 40 µg of lysates were loaded into 10% TGX stain-free gels (Bio-Rad) for denaturing sodium dodecyl sulfate (SDS) polyacrylamide gel electrophoresis (PAGE). Migrated proteins were transferred onto a PVDF membranes using the Trans-Blot turbo (Bio-Rad). Membranes were blocked using 5% milk in TBST before being incubated overnight with primary antibody. The following antibodies and dilutions were used: NRP2 (Thermo Fisher, Waltham, MA, USA; PA5-75451) 1:1000, Furin (Thermo Fisher; PA5-96680) 1:1000, SREBP-2 (Thermo Fisher; PA5-88943) 1:2500, SARS-CoV-2 spike S10 (1A9) (GeneTex; GTX632604) 1:1000. Membranes were then incubated with Jackson ImmunoResearch secondary donkey-anti-rabbit (1115-035-152) or goat anti-mouse (1115-035-062) antibody conjugated with horseradish peroxidase depending on the identity of the primary antibody, at a 1:20,000 dilution. Membranes were blotted using clarity ECL solution reagent (Bio-Rad) and imaged on the ChemiDoc MP (Bio-Rad). Blot images were cropped and adjusted using Image Lab (Bio-Rad).

### 2.7. RNA Extraction and Quantitative Real-Time PCR (RT-qPCR) for HCoV-229E and Host Genes

Total RNA was extracted from cells using the RNeasyPluskit (Qiagen, Hilden, Germany) and quantified using a NanoDrop (Thermo Fisher Scientific). A total of 400 ng of isolated RNA was reverse transcribed into cDNA using the iScript Reverse Transcription kit according to the manufacturer’s protocol. qPCR was performed using SSOAdvanced Universal SYBR GreenSupermix (Bio-Rad) according to the manufacturer’s instructions. Primers specific to the HCoV-229E E protein were used with the following sequences, FWD 5′-TGGCCCCATTAAAAATGTGT-3′ and REV 5′-CCTGAACACCTGAAGCAAT-3′. Primers were present at a final concentration of 250 nM in a total volume of 10 µL. CFX Connect Real-Time PCR Detection System (Bio-Rad, Hercules, CA, USA) was utilized for the analysis of the qPCR. We used the 2^−∆∆Ct^ to calculate the relative levels of mRNA and fold changes between conditions. GAPDH was used to normalize samples.

### 2.8. Viral RNA Extraction and Quantitative Real-Time PCR (RT-qPCR) for SARS-CoV-2

Cellular RNA was extracted using the Qiagen© RNeasy Mini kit (Qiagen) according to the manufacturer’s instructions. Viral RNA (vRNA) was extracted from the supernatant using the Qiagen© QIAamp Viral RNA Mini Kit (Qiagen). All cellular qRT-PCR was performed in triplicate on cDNA synthesized as previously described [54]. vRNA was quantified by Qiagen© Quanti-fast RT probe master mix (Qiagen) using primer/probe sets specific for the SARS-CoV-2 subgenomic E gene. The reactions were performed on a StepOnePlusTM Real-Time PCR System in a 96-well plate (Thermo Fisher) as previously described [55], in accordance with MIQE guidelines [56].

### 2.9. Bioinformatic (miRDB and Gene Ontology) and Hypothesis-Driven Approach in Selecting Targets

A miRNA target prediction tool, miRDB [29], was used to determine predicted targets of miR-24-3p. The miRDB tool combines data from large-scale RNA-seq data (overexpressing 25 different miRNAs) along with direct miRNA:mRNA interactions determined through crosslinking immunoprecipitation (CLIP). The authors then trained a support vector machine (SVM) model, termed MirTarget, on these datasets to generate a prediction score, with those over 50 being considered predicted targets [29]. We conducted a search for miR-24-3p in miRDB, which yielded 959 targets scoring between 50 and 99. Targets with a score over 90 were chosen for the initial analysis. For subsequent analyses, targets were filtered by expression levels within each cell line studied as well as miRDB scores over 50, and those that were highly expressed (RPKM > 20) were included in the Pathway Gene Ontology. Certain targets were taken from these lists in a hypothesis-based approach after cross-referencing with the available literature.

### 2.10. Statistical Analysis

Data are presented as the mean of replicates, with error bars representing the standard error of the mean. Statistical significance was evaluated using unpaired two-tailed Student’s *t*-test on GraphPad Prism 10.1.1, and *p*-values less than 0.05, 0.01, 0.001, or 0.0001 were deemed significant and labeled accordingly.

## 3. Results

### 3.1. miR-24-3p Inhibits SARS-CoV-2 Infection In Vitro

To study the effect of miR-24-3p on SARS-CoV-2 infection, Calu-3 lung cells were reverse transfected with miR-24-3p for 24 h before infecting with SARS-CoV-2 for either 24 h, 48 h, or 72 h followed by quantifying 50% tissue culture infectious dose (TCID50) through RT-qPCR for the viral RNA (vRNA) (Figure 1A). Following infection, supernatant was collected separately to assess produced virions (Figure 1B), while the cells were collected and lysed to probe for viral replication (Figure 1C). Compared to the control scramble miRNA (con-miR), miR-24-3p led to significant decreases in both viral replication (intracellular vRNA) and released virions (supernatant vRNA) at all time points assayed between 24 and 72 h. Interestingly, the decrease in supernatant, or extracellular vRNA, was more pronounced than intracellular vRNA; this may be due to the lag time between replication and viral assembly and release or it may indicate additional inhibition at virion release. We also confirmed that miR-24-3p and con-miR were not cytotoxic by MTT assay (Appendix A). The findings suggest that miR-24-3p is antiviral against SARS-CoV-2 by inhibiting at least one, but potentially more, stages of viral infection given that both intracellular and extracellular vRNA was reduced.

### 3.2. Predicting Antiviral Effectors of miR-24-3p Using Bioinformatic Tools

Given that miR-24-3p inhibited the viral infection of SARS-CoV-2 in lung cells, we sought to deduce the antiviral mechanism of action. miRNAs generally have numerous targets [57], which can make deducing specific mechanisms of action challenging. To elucidate possible effectors, several bioinformatic resources for predicting miRNA targets have been generated to date [58], including miRanda [59], TargetScan [60], and miRDB [29]. Numerous targets of miR-24-3p were identified using miRDB; therefore, we narrowed down the list to select targets that are effectors promoting infection for SARS-CoV-2 or other viruses (Table 1). In addition, we also found four separate seed sites for miR-24-3p in SREBP2, for which there is evidence it contributes to infection of HCV [14,49] and SARS-CoV-2 [17].

To assess the impacts of miR-24-3p more broadly, we subjected the highly expressed (RPKM ≥ 20) [29] mRNA targets to Panther Pathway Gene Ontology (GO) [61,62] for three commonly used cell lines in studying SARS-CoV-2: Calu-3, Huh7, and A549s (Figure 2). Highly expressed targets were assayed given that target expression is an important aspect of miRNA mechanisms [10]. We also performed GO on the highest scored (≥90) miRDB predicted targets regardless of cell line expression (Appendix A) to see if any important processes were overlooked when filtering by expression.

Combining takeaways from the hypothesis-driven approach (Table 1) with broader gene ontology (GO) suggests that the inhibition of viral entry may be a key mechanism employed by miR-24-3p against SARS-CoV-2. This is exemplified through NRP1 and NRP2, furin, and SREBP2 (Table 1), which play pivotal roles in the entry of SARS-CoV-2 and other viruses [14,17,23,33,36,37,39,40,41,44,45,46,47,63,64]. Scavenger receptor class B member 1 (SR-B1 or SCARB1), a receptor for high density lipoproteins (HDLs) was also identified, which acts as a coreceptor for SARS-CoV-2 [65] and has been explored as an antiviral target [51,66]. Moreover, additional targets such as interferon gamma (IFNγ) and PTGER4, which modulates interferon response [67], may indirectly affect SARS-CoV-2 entry through the interferon regulation of the SARS-CoV-2 entry receptor, angiotensin converting enzyme 2 (ACE2) [68]. This mechanism of indirect regulation is supported by the related interleukin signaling pathway (P00036) and angiogenesis (P00005) pathways identified by GO (Figure 2). It is worth noting, there were also several poorly characterized infection-promoting plasma membrane protein targets, like CDH7 [69], which may contribute to viral entry. All miRNA:mRNA interactions in the 3′-UTR for each of these targets in Table 1 were also generated, along with the predicted binding energies (Appendix A).
viruses-16-01844-t001_Table 1Table 1Selected predicted targets of miR-24-3p that promote viral infection. miRDB was used to generate a list of predicted targets of miR-24-3p. Targets were then selected in a hypothesis-driven manner for those that have previously been implicated in SARS-CoV-2 pathogenesis or another virus. A target was deemed to promote viral infection if the authors performed mechanistic experiments showing that the protein clearly contributed to viral proliferation or if targeting the gene (ex: CRISPR-KO, siRNA, small molecule inhibitors, etc.) led to a decrease in viral proliferation. If the target was identified as differentially regulated at a protein or RNA level in a high-throughput analysis but was not explored further, it was labeled accordingly as either upregulated or downregulated.Title 1VirusPromote Viral Infection or UpregulatedReferenceNRP1SARS-CoV-2Promote infection[39]HTLV-1Promote infection[70]NRP2LUJVPromote infection[45]SARS-CoV-2Promote infection[47]HCMVPromote infection[44]HCMVPromote infection[45]SARS-CoV-2; IAVUpregulated[42]FURINSARS-CoV-2Promote infection[35]IAVPromote infection[33]HIVPromote infection[32]SR-B1HCVPromote infection[71]SARS-CoV-2Promote infection[64]SARS-CoV-2Promote infection[65]SREBP2SARS-CoV-2Promote infection[17]HCVPromote infection[49]HCVPromote infection[14]PTGER4LCMVPromote infection[66]RABVUpregulated[72]BEFVPromote infection[73]CDH7MERS-CoV; HCoV-229EPromote infection[68]TOP1EBOVPromote infection[74]EV71Promote infection[75]HBVPromote infection[76]HIVPromote infection[77]

### 3.3. miR-24-3p Downregulates Predicted Targets Related to Viral Entry

We next sought to validate the entry-related effectors, NRP1, NRP2, furin, and SREBP2, which have been mechanistically well-studied in the context of SARS-CoV-2 [16,30,34,35,36,38,39,40,41,46]. To determine if miR-24-3p downregulates these targets, we performed RT-qPCR for furin (Figure 3A) and SREBP2 (Figure 3B) on cell lysates following reverse transfection with miR-24-3p. Both furin and SREBP2 were significantly downregulated by miR-24-3p at the mRNA level. NRP1 has previously been experimentally validated elsewhere as a miR-24-3p target using a 3′ UTR luciferase assay [48], and shares a high homology with NRP2, suggesting that it is also likely targeted directly by miR-24-3p. These findings suggest that miR-24-3p directly targets important cell entry factors of SARS-CoV-2.

### 3.4. SARS-CoV-2 S Protein Pseudoviruses Experience Reduced Viral Entry Following miR-24-3p Treatment

To delineate the cell entry step of the SARS-CoV-2 life cycle and study the mechanistic effects of the miR-24-3p targets, we generated the SARS-CoV-2 S pseudovirus with a luciferase reporter to quantify viral entry (Appendix A) [35,53]. If entry is impacted by miR-24-3p, a quantifiable decrease in luciferase signal is expected compared to control. Pseudotyped virions were generated by transfecting HEK293T cells with a plasmid encoding the SARS-CoV-2 S glycoprotein, along with pNL4 3.Luc.R E (Figure 4A) [35,53]. The cells successfully produced the virions evidenced by the structural HIV-p24 as well as the SARS-CoV-2 S protein detected by western blot in the extracellular media (Figure 4B).

Supporting our hypothesis that SARS-CoV-2 viral entry is impacted by miR-24-3p, pre-treatment with miR-24-3p led to a significant decrease in pseudovirus infection (measured by luciferase intensity) in both Huh7 and Calu-3 cells when compared with a control miRNA mimic (Figure 4C,D). These findings indicate that miR-24-3p inhibits the entry of SARS-CoV-2.

### 3.5. miR-24-3p Maintains Antiviral Effectiveness in Response to Common SARS-CoV-2 Mutants D614G and N501Y

A major factor that makes SARS-CoV-2 an effective virus and therefore, extremely difficult to target with the current vaccines, monoclonal antibody treatments and antiviral therapies, is the ability of SARS-CoV-2 to mutate into more resistant variants [78]. Given that miRNAs target multiple mRNAs, it is possible that they may be more resistant to escape mutants. Simultaneously targeting multiple proviral proteins has previously been demonstrated with the success of antiviral inhibitor cocktails applied for treating HCV [79] and HIV [80]. Furthermore, there is evidence that targeting host proteins rather than viral proteins with antiviral therapies is more resistant to viral mutagenesis and escape mutants (Reviewed in Ji & Li., 2020) [81].

The D614G S mutation has been associated with increased viral fitness in SARS-CoV-2, evidenced by its presence in all variants that have emerged since the initial outbreak [6,78]. Interestingly, D614G has a mutation near the furin cleavage site, which has been demonstrated to improve furin cleavage [82]. The improved cleavage and its additional tendency to favor the receptor binding domain (RBD) in the “up” conformation have, therefore, been suggested as reasons for the improved infectivity of D614G [82]. N501Y is another common mutation, located further upstream of the furin site in the RBD, and has been shown to increase the affinity of the RBD for the ACE2 entry receptor [83]. Interestingly, this mutation is found in most SARS-CoV-2 variants of concern, including the alpha (B.1.1.7), beta (B.1.351), gamma (B.1.1.28) and omicron strains (BA.1, BA.2, BA.2.12.1, BA.4 and BA.5) [78].

To assess the versatility of miR-24-3p in conferring protection from infection, we assessed the entry of SARS-CoV-2 D614G and N501Y mutants in Huh7 (Figure 5A) and A549 cells stably expressing ACE2 (Figure 5B). Despite the improved fitness of these variants, miR-24-3p maintained a high degree of entry inhibition. Therefore, miR-24-3p exhibited a degree of resistance to highly prevalent S mutations.

### 3.6. miR-24-3p Downregulates Furin, NRP2 and SREBP2 While Reducing Virion Production and Infectivity

With evidence that miR-24-3p is antiviral against SARS-CoV-2, downregulates key entry factors at an mRNA level, and plays a role in inhibiting entry specifically, we sought to explore the mechanism of this inhibition further. While NRP1 and NRP2 have clear roles acting as coreceptors during entry [39,40,41] and SREBP2 induces cholesterol biosynthesis [51], which binds to S to enhance entry and may play a role in viral production [64], furin-mediated cleavage most frequently occurs on the golgi bodies during viral production [31,35]. Furin primes the virus by cleaving the S1–S2 subunits, via the polybasic R-R-A-R motif during viral production, leading to improved entry when these primed virions encounter a target cell [31,35].

To study the role of miR-24-3p in production, and thus, the effect of furin and SREBP2 knockdown in virus-producing cells, HEK293T cells were reverse transfected with miR-24-3p the day before transfection with the pseudovirus plasmids (S glycoprotein and pNL4-3.Luc.R-E) to produce infectious pseudovirions (Figure 6A). Because D614G seems to rely more heavily on furin cleavage [82] and results in more infectious virions, we used this variant for pseudovirus treatments. Lysates from the pseudovirus-producing HEK293Ts (Figure 6B) and an aliquot from extracellular media (Figure 6C) were analyzed via western blotting. In this experiment, S in lysate was solely dependent on the transfection efficiency of the S glycoprotein plasmid; conversely, extracellular S would be a gauge of pseudovirus production and release because there is no alternative method for S to exit the cytosol. We also performed densitometry for these blots (Appendix A). The S mRNA does have a predicted miR-24-3p seed binding site [84]; however, no decrease in intracellular S was observed (Figure 6B), suggesting miR-24-3p exhibits inhibitory effects on host factors instead. Nonetheless, there was a robust decrease in extracellular S (Figure 6C) and there was substantially reduced infectivity when extracellular media (containing pseudovirus) from miR-24-3p pre-treated cells was applied to healthy cells (Figure 6D). Together these findings suggest virion release is impaired by miR-24-3p, and this may be in part due to the SREBP2 downregulation.

In addition, we also noted a decrease in S2′ abundance (Figure 6B). Given that S2′ cleavage requires previous cleavage of S1-S2 [31], furin activity may be impaired. This hypothesis was supported by the findings of decreased infectivity when normalizing the infectivity of miR-24-3p primed pseudovirus by the amount of extracellular S detected (Figure 6E).

To confirm that the miR-24-3p downregulation occurs at the protein level in addition to the mRNA level, we probed the pseudovirus-producing cells for furin, NRP2, and SREBP2. As expected, cell lysates had decreased the abundance of these proviral host proteins involved in SARS-CoV-2 entry (Figure 6B). The antiviral effects of downregulating these proteins, particularly furin [31,35,63], on SARS-CoV-2 entry and entry of other viruses has been well-characterized [14,17,44,45,46,47], strengthening our findings. Taken together, the findings indicate an antiviral mechanism targeting virion production and producing less infectious virions, supporting the live virus data. In addition, we show that the previously validated factors involved in the entry of the virus are downregulated at the protein level by miR-24-3p.

### 3.7. The Antiviral Effects of miR-24-3p Are Reduced for Human Coronavirus 229E Compared to SARS-CoV-2

In the coronavirdae family, the presence of a furin cleavage site varies with notable members like MERS-CoV and SARS-CoV-2 having the site while SARS-CoV and HCov-229E do not [31]. This difference is also observed in SARS-CoV-2 utilizing NRP1 and having a different entry receptor than HCoV-229E or MERS-CoV [2]; accordingly, inhibition of NRP1/ACE2 has a very minimal effect on HCoV-229E viral titre [85]. Therefore, these differences make HCoV-229E an ideal coronavirus to test the importance of NRP1/2 and furin downregulation in the antiviral mechanism of miR-24-3p. In line with our previous findings, we determined that miR-24-3p led to a decrease in HCoV-229E vRNA that was not statistically significant (*p* = 0.29) (Appendix A). vRNA was normalized to GAPDH, which remained stable during miR-24-3p and con-miR treatment followed by HCoV-229E infection (Appendix A). These findings suggest that the regulation of other identified factors such as SREBP2, TOP1, and SR-B1 may result in some downregulation but that NRP1/2 and furin make a relatively larger contribution to the antiviral mechanism of miR-24-3p.

## 4. Discussion

Popular vaccines SpikeVax (Formally, mRNA-1273) and BTN162b2 [5], along with antiviral treatments like Paxlovid [86] (nirmatrelvir; ritonavir), and monoclonal antibodies [69] reflect the concerted global effort to combat SARS-CoV-2 from 2019 to present. Unfortunately, however, along with all the successful approaches to target SARS-CoV-2, there are still limitations with the current therapeutics. For instance, escape mutants have made vaccine development a constant challenge since early in the COVID-19 pandemic [5] and, more recently, the generation of escape mutants against nirmatrelvir has been demonstrated experimentally [7]. Furthermore, Paxlovid is most effective for mild to moderate SARS-CoV-2 infections and may have several significant drug interactions with other medications [86], which can pose a problem for the higher risk, older population.

In the present study, we demonstrate that miR-24-3p is antiviral against SARS-CoV-2 (Figure 1) as well as SARS-CoV-2 S pseudovirions (Figure 4) and pseudovirus variants (Figure 5). We also note this effect is limited when compared to the more distantly related betacoronavirus, HCoV-229E (Appendix A), due to targeting specific host factors important in SARS-CoV-2 entry but not required by HCoV-229E. miRNA-based and miRNA targeting therapies have potential as antiviral treatments with the benefit that they can target multiple pathogenic factors simultaneously, which mirrors the successful multidrug regimens applied to HCV [78] and HIV [79]. Multitarget strategies are particularly useful at minimizing escape mutants; accordingly, miR-24-3p effectiveness was maintained against SARS-CoV-2 S variants (Figure 5). Furthermore, miRNAs or targeting miRNAs can be a successful antiviral therapy, such as the example of the anti-mer against miR-122 (to treat HCV), Miravirsen, which successfully completed phase II clinical trials (NCT01200420) [18,19]. Traditionally, the cost of synthesizing these drugs has been a major limiting factor; however, emerging enzymatic synthesis techniques are drastically improving the cost effectiveness and scale of synthesis [87].

Given the high degree of predicted and known targets of miRNAs, predicting the specific mechanism(s) of regulation during pathology can sometimes be arduous. To further pinpoint the mechanisms for the observed inhibition of SARS-CoV-2 by miR-24-3p, we employed bioinformatic methods combined with a hypothesis-driven approach (Table 1, Figure 2 and Appendix A). Several predicted miR-24-3p targets were identified across various cellular processes that are often manipulated by viral pathogens. Interestingly, angiogenesis and interleukin signaling were identified. ACE2 expression directly influences angiogenesis [88] and is the entry receptor for SARS-CoV-2 [2]. Downregulation of ACE2 expression is an antiviral mechanism of miR-185 [17], suggesting that this may also be targeted by miR-24-3p [84]. Moreover, interleukin signaling is reasonable given findings that the related IFNγ signaling can increase ACE2 expression, furthering SARS-CoV-2 pathogenesis [68]. When looking at specific predicted targets, this hypothesis seems plausible considering that IFNγ was a highly predicted target (Table 1) and IFNγ is also modulated by another predicted target, PTGER4 [67].

We show that in targeting the production of SARS-CoV-2, miR-24-3p leads to a downregulation of furin and SREBP2 at the mRNA (Figure 3) and protein level (Figure 6). Furthermore, miR-24-3p has been shown to downregulate NRP1 [48], and we show that miR-24-3p downregulates the protein abundance of NRP2 (Figure 6). NRP1 and NRP2 have been shown to promote the entry of a broad range of viruses by acting as coreceptors [39,40,44,45,46,47], while SREBP2 likely affects entry through cholesterol biosynthesis and modulation of lipid intermediates [49,65]. Cholesterol promotes SARS-CoV-2 entry through association SR-B1 [65], which was also identified as a miR-24-3p target (Table 1) but was not explored experimentally in the present study.

In addition to entry at recipient cells, miR-24-3p also affects the production of new virions, evidenced by fewer SARS-CoV-2 virions produced (Figure 1) and S pseudovirions produced (Figure 6) following miR-24-3p pre-treatment. Moreover, the produced virions were significantly less infectious per virion (Figure 6). We suggest that furin and SREBP2 downregulation (Figure 3 and Figure 6) likely contribute to this through reduced cleavage of the S1/S2 furin site and differential regulation of cholesterol and lipids in the viral envelope through SREBP2 downregulation (Figure 7). Supporting the decreased infectivity is the finding that in miR-24-3p treated cells producing S pseudovirions (Figure 6), S2′ abundance is decreased, which requires preliminary S1-S2 cleavage by furin [31]. Interestingly, however, S2 levels did not seem affected by the treatment, which could be due to the experimental setup or the presence of trypsin, which is known to cleave the polybasic furin site [71]. The effects of downregulation of furin with siRNA [35] and inhibitors [31,35] against SARS-CoV-2, however, is already very well characterized (reviewed in Osman et al., 2022) [63].

Although furin is known to play a large role in SARS-CoV-2, it is important to consider the contribution of SREBP2 as well. HCV, which is also a + sense RNA virus, has a high cholesterol content in its viral envelope [89] and is also dependent on SR-B1 [90], suggesting the downregulation of SREBP2 could be another major contributor to the antiviral effects against SARS-CoV-2 during virion production. Given that miR-24-3p is also antiviral against HCV [22] and miR-185 likely inhibits SARS-CoV-2 through a similar mechanism [17], this pathway could have important implications for broadly treating RNA virus infections.

While all of these factors are enticing targets for therapy and potential multi-target strategies, it is important to address the limitations of this approach. We previously describe the benefits of targeting host factors as antiviral therapies [81]; however, there is also the drawback that, unlike viral proteins, host proteins often serve physiological roles in normal homeostasis. For instance, NRP1/2 play roles in regulating vascular endothelial growth factor (VEGF) governing angiogenesis [43]; furin is important for processing transformation growth factor β1 (TGF-β1) controlling cell proliferation and differentiation [91]; and SREBP2 is a key modulator in canonical cholesterol homeostasis [49]. It should be noted, however, that there have been promising phase IIb safety profiles for an immunotherapy targeting furin (NCT02346747) [92], suggesting that with proper dosing, targeting these host factors could still be an effective therapy.

Overall, we demonstrate that miR-24-3p is antiviral against SARS-CoV-2, inhibiting both the full replicative virus and a model system for the isolated study of viral entry. The effectiveness of the miRNA treatment was resistant to common SARS-CoV-2 S mutations and further mechanistic insights were explored, highlighting that furin, NRP1/2, and SREBP2 play a key role in the attenuation of entry. The findings demonstrate the benefits of using miRNAs as therapeutics and suggest that these targets may be combined in already available therapies to improve upon viral inhibition as issues with SARS-CoV-2 persist and we look forward, we must consider new approaches to antagonize current strains and, more importantly, coronaviruses that are likely to emerge in the coming decades.

## Figures and Tables

**Figure 1 viruses-16-01844-f001:**
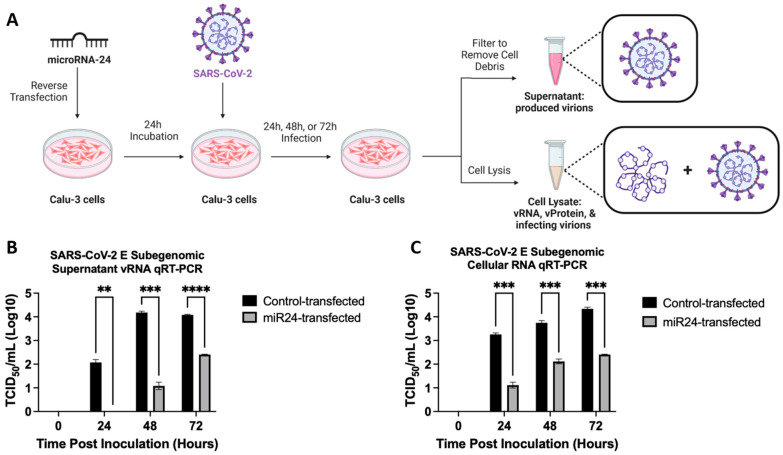
miR-24-3p inhibits SARS-CoV-2 replication and virion production. (**A**) Calu3 Cells were reverse transfected with miR-24-3p or con-miR for 24 h prior to infection with SARS-CoV-2, MOI of 0.01, for 24 h, 48 h, or 72 h. RT qPCR (technical triplicate) was then performed to quantify relative vRNA in samples. (**B**) Production of SARS-CoV-2 was assessed by collecting an aliquot of the supernatant at each time point. (**C**) Subgenomic cellular RNA of SARS-CoV-2 was assessed by lysing cells at each time point. *n* = 3. Error bars represent SEM. *p* < 0.01 **, 0.001 ***, 0.0001 ****.

**Figure 2 viruses-16-01844-f002:**
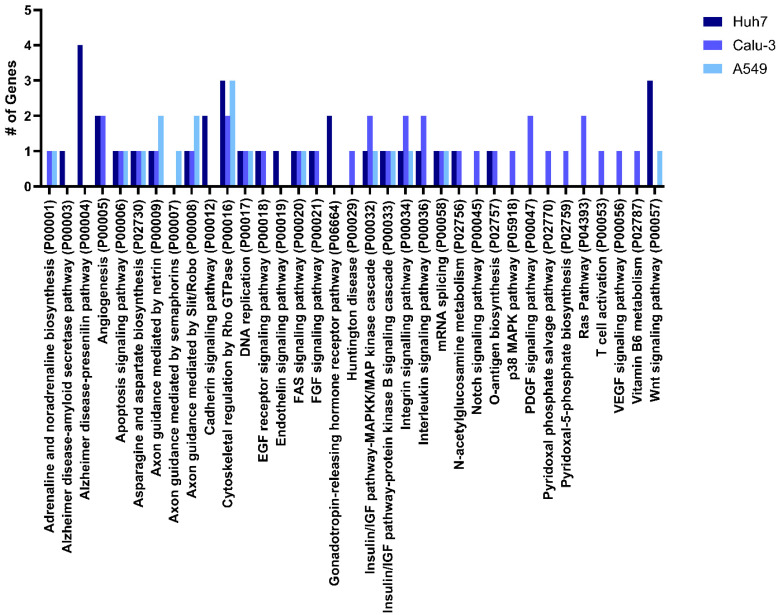
Pathway Gene Ontology (GO) of highly expressed miR-24-3p targets. Panther GO was performed on miRDB predicted targets that are highly expressed (RPKM ≥ 20) in the cell lines used in the present study: Huh7, Calu-3, and A549. Most genes belong to unclassified categories; however, these were removed for clarity. A total of 75–102 targets were highly expressed in each cell line.

**Figure 3 viruses-16-01844-f003:**
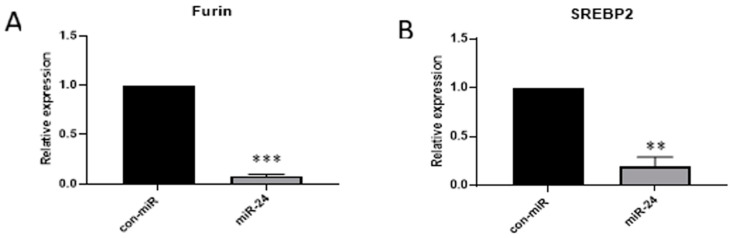
miR-24-3p downregulates Furin and SREBP2 mRNA. Calu-3 cells were reverse transfected with 100 nM miR-24-3p or con-miR for 72 h before lysis for RT-qPCR. Treatment with miR-24-3p post-transcriptionally represses (**A**) furin and (**B**) SREBP2 at the mRNA level. *n* = 3. Error bars represent SEM. *p* < 0.01 **, 0.001 ***.

**Figure 4 viruses-16-01844-f004:**
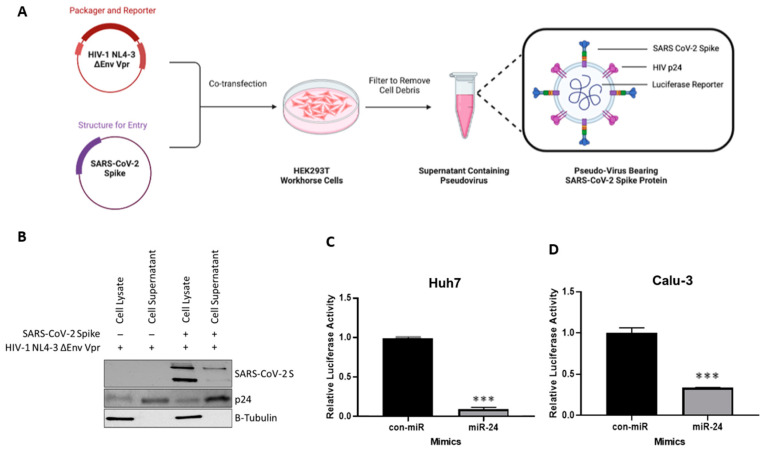
miR-24-3p decreases the entry of SARS-CoV-2 in an S pseudovirus model. (**A**) Scheme depicting the generation of SARS-CoV-2 S pseudotyped virus. (**B**) Validation of pseudovirus components via western blot of HEK293T producing cell lysates and extracellular supernatant. (**C**) Pseudovirus entry assay quantified by luciferase microplate reader (technical triplicate). Assay was performed after 24 h reverse transfection of miR-24-3p followed by 48 h pseudovirus infection and lysis using a passive lysis buffer in either (**C**) Huh7 or (**D**) Calu-3 cells. Both con-miR and miR-24-3p values were normalized to the average con-mR value. *n* = 3. Error bars represent SEM. *p* < 0.001 ***.

**Figure 5 viruses-16-01844-f005:**
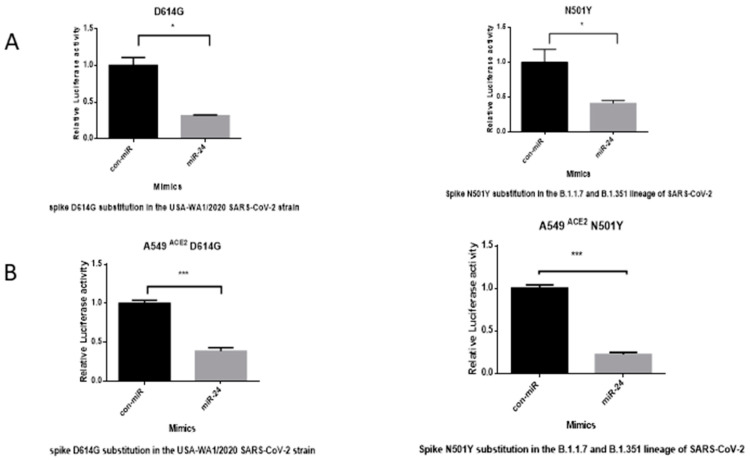
miR-24-3p maintains effectiveness against common SARS-CoV-2 S mutations. Pseudovirus entry assay quantified by luciferase microplate reader in technical triplicate. The assay was performed after 24 h reverse transfection of miR-24-3p or con-miR followed by 48 h pseudovirus S infection with D614G or N501Y S mutants in either (**A**) Huh7 or (**B**) ACE2 stably expressing A549 cell line. Lysis was performed using a passive lysis buffer. Both con-miR and miR-24-3p values were normalized to the average con-miR value. *n* = 3. Error bars represent SEM. *p* < 0.05 *, 0.001 ***.

**Figure 6 viruses-16-01844-f006:**
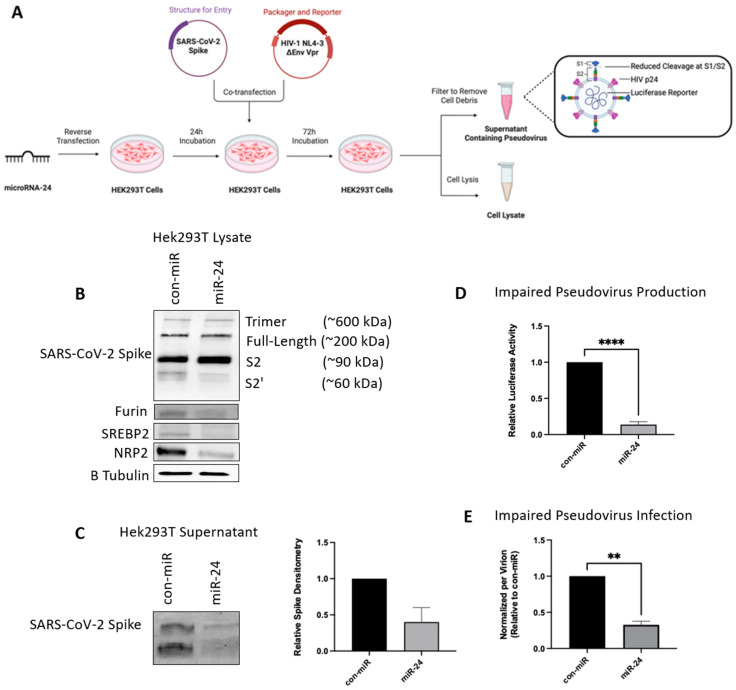
miR-24-3p downregulates Furin, SREBP2, and NRP2 impairing production and entry of SARS-CoV-2. (**A**) Scheme depicting experimental workflow for assessing target abundance and pseudovirus S production during miRNA pre-treatment. Briefly, HEK293T cells were pre-treated with miR-24-3p or con-miR 24 h before transfection with the plasmids to produce pseudovirions. After 48 h, the pseudovirus was collected and the pseudovirus produced during miR-24-3p treatment or con-miR treatment were then used to infect healthy untreated Huh7 cells. A luciferase assay was then performed on these Huh7 cells to quantify the amount of pseudovirus produced. (**B**) Western blot analysis of lysates from control or miR-24-3p-treated HEK293Ts producing S-pseudovirus. Several essential proviral targets and the viral S protein were probed for. (**C**) Western blot analysis of supernatant from control or miR-24-3p-treated HEK293Ts cells producing S-pseudovirus. (**D**) S-pseudovirus entry assay performed on non-treated Huh7 cells following production in HEK293T cells reverse transfected with miR-24-3p or con-miR. (**E**) S-pseudovirus entry assay from (**C**) normalized to total S production from (**D**). *n* = 2. Error bars represent SEM. For the luciferase data, both con-miR and miR-24-3p values were normalized to the average con-mR value. *p* < 0.01 **, 0.0001 ****.

**Figure 7 viruses-16-01844-f007:**
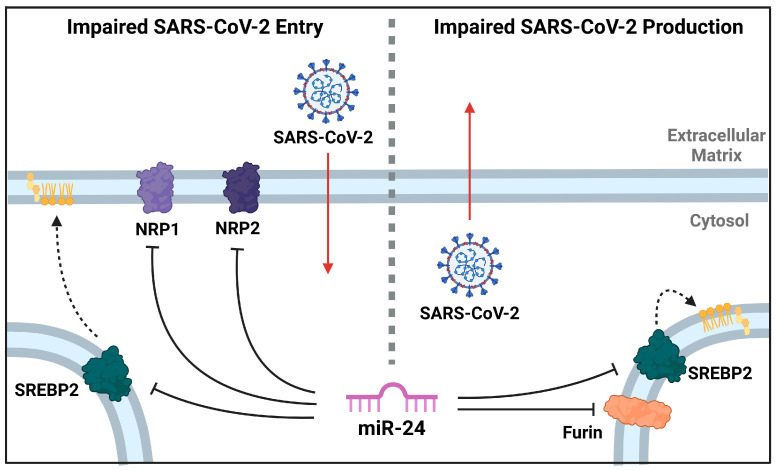
Diagram illustrating the targets and effects of miR-24-3p during SARS-CoV-2 infection.

## Data Availability

All data referenced in this study are included in the article and Appendix A. Further inquiries can be directed to the corresponding author.

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
