# Peer review of "miR-24-3p Is Antiviral Against SARS-CoV-2 by Downregulating Critical Host Entry Factors"

_viruses, 2024, doi:10.3390/v16121844_

Round 1
Reviewer 1 Report
Comments and Suggestions for Authors
Review of manuscript "miR-24 is antiviral against SARS-CoV-2 by downregulating critical host entry factors" by Evers et. al for consideration in Viruses.
In this manuscript, the authors have analyzed host factors that are putatively regulated by microRNA 24 to study their role as antiviral factors for SARS-CoV-2 infection. This is well designed study and well written manuscript as well; clearly outlining the workflows used and explaining the data obtained. Considering the myriad roles of miRNAs in infectious processes, studies like these can help identify novel intervention targets for important human and animal viral pathogens. Technically the manuscript is sound; however a -nListed below are my comments on the manuscript.
Major comments on the manuscript:
1. Line 31, provide references.
2. The first line of the abstract raises a very interesting point that SARS-CoV-2 has moved from a pandemic to an endemic situation. This is better moved to the introduction and should be referenced.
3.Lines 71-103 are better suited in a discussion in context with the results in the manuscript.
4.In line 125, please clarify which lineage this SARS-CoV-2 belongs to, the original Wuhan or delta / lambda or omicron ?
5. Line 137, the manufacturer should be Agilent instead of Aligent. Were mutations confirmed by sequencing ?
6. In Line 155, did the authors do dose response studies. Are they transfecting miR-24-5p or miR-24-3p ? This should be made crystal clear throughout the manuscript.
7. Line 165 A549ACE2 should be A549-ACE2 cells. Please refer where these cells were obtained from or if they were generated in the lab and how.
8. Were miR-24 expression levels determined in untransfected cells prior to infection ? And following infection ? Is miR-24 deregulated by SARS-CoV-2 infection ?
9. Add Calu3 to Line 165 as well; else that reference appears out of place while reading the manuscript.
10. Line 181, declare what DC assay is ? Was it a denaturing or native PAGE ?
11. Please include details of primary and secondary antibodies (isotype, manufacturer and dilution used) in the methods to allow reproducibility.
12. Line 200. Have the authors verified that GAPDH levels are unaffected by transfection / SARS-CoV-2 infection ?
13. Line 204, declare viral RNA before using vRNA. Please state cycling parameters to ensure reproducibility.
14. Line 209, were viral genome equivalents determined by quantitative PCR with standard curves. If yes, please include statement that MIQE guidelines were adhered to.
15. Line 213, please include a statement on how miRDB assigns scores to predicted targets. In line 215, include a rationale for choosing 50 as a cutoff.
16.Line 220, unaired should be unpaired. In my opinion, the correct statistical method here would be a one-way/ two-way ANOVA with post-hoc Geisser correction. Please state version of GraphPad Prism used.
17. Results section 3.1 only describes what was done but does not discuss the data that were obtained in Figure 1. The authors should briefly state the objective of the section and then elaborate on the results obtained.
18. Figure 1B Legend miR24 should be miR-24-**(the 5p or 3p miRNA used).
19. Line 243, did the authors check if miR-24 can directly target the SARS-CoV-2 genome ?
20. In Figure 3, please show the mRNA UTR on top and the miRNA below it. Show Watson-Crick pairs with straight lines and non-WC pairs with colon symbols. Indicate nucleotide positions in the 3'-UTR that are targeted. Make a graphic / table showing all mRNA: miRNA alignments ; the authors can sort them in descending stability based on free energy calculations.
21. For controls, please show error bars. Please replace miR-24 with miR-24-3p since that seems to be the miRNA studied in the manuscript.
Have the authors considered the effect of miR-24-3p pretreatment on the cell cycle? Since miRNA transfections raise endogenous miRNA levels to non-physiological levels, please check for cytotoxicity and include that as supplementary data. miR-24-3p is a known regulator of cell cycle and this may alter viral replication. For many viruses, a particular stage of the cell cycle is preferred for infection. The reduction in viral titers could be due to alteration in cell cycle as an additional mechanism. The statement in Line 315-316 is only partly substantiated by the data. Have the authors considered infecting the cells and then transfecting with mimic transfection ? Or treat cells with anti-ACE2 or anti-SARS2 antisera in miR-24 transfected cells ? If entry is the key step regulated by miR-24, the authors would see no difference when cells are infected prior to transfection. Additionally, ACE2 is a key receptor for SARS-CoV-2 along with TMPRSS2. Does miR-24 mimic transfection prior to infection alter ACE2 / TMPRSS2 transcript / protein expression? Did the authors validate overexpression of miR-24 in transfected cells by qPCR relative to a housekeeping gene ?
22. Did the authors titer the pseudovirus in Figure 4 and what MOI infections were done in Figure 4,
23. Line 342, there is a typo ACE2 is spelled as AEC2.
24. Data from Figure 5A does not demonstrate increased fitness of the D614G or the N501Y mutants compared to wild type virus in Huh7 cells. Whereas the wild type virus replicates well and is much more significantly inhibited in Huh7 cells, the degree of inhibition for the mutant viruses is at best modest. The effect is certainly more pronounced in A549 cells stably expressing human ACE2. The authors should also compare the level of miR-24 inhibition between D614G and the N501Y relative to controls using one-way ANOVA.
25. Please use either con-miR or Con-mir consistently throughout the manuscript. Did the authors titrate the pseudovirus supernatants before infection ? What was the effective MOI or TCID50 for these experiments ?
26. Line 454, antimere is a typo of antimer.
Author Response
Reviewer #1:
Comments and Suggestions for Authors
Review of manuscript "miR-24 is antiviral against SARS-CoV-2 by downregulating critical host entry factors" by Evers et. al for consideration in Viruses.
In this manuscript, the authors have analyzed host factors that are putatively regulated by microRNA 24 to study their role as antiviral factors for SARS-CoV-2 infection. This is well designed study and well written manuscript as well; clearly outlining the workflows used and explaining the data obtained. Considering the myriad roles of miRNAs in infectious processes, studies like these can help identify novel intervention targets for important human and animal viral pathogens. Technically the manuscript is sound; however a -n Listed below are my comments on the manuscript.
Major comments on the manuscript:
Comment 1. Line 31, provide references.
Response 1: A reference for this has been provided. For details, see the next response.
Comment 2. The first line of the abstract raises a very interesting point that SARS-CoV-2 has moved from a pandemic to an endemic situation. This is better moved to the introduction and should be referenced.
Response 2: While it is well-accepted that SARS-CoV-2/COVID-19 will eventually become endemic like previous pandemics of influenza or HIV, few research articles have been bold enough to confirm the world is currently in that stage. Experts, like Aron Hall heading the Centre for Disease Control (CDC) of the USA, however, have stated that it is currently endemic in an interview with NPR (Aug. 9, 2024; Rob Stein). With this in mind we have decided to soften our language to include that the shift to “endemic is expected and beginning to be modeled” with a reference to this in the introduction.
We agree with the reviewer on moving this information to the introduction rather than including in both the abstract and introduction. The first sentence in the introduction has been re-written as follows:
“Despite all the progress in treating SARS-CoV-2, escape mutants to current therapies remain a constant concern.”
Comment 3. Lines 71-103 are better suited in a discussion in context with the results in the manuscript.
Response 3: When preparing the manuscript, we originally included this information within the results to help explain the reasoning for the experiments performed in figure 6 and 7; however, this reduced the clarity and succinctness of the text and thus, we attempted to move it to the discussion. The issue with this text being in the discussion was that, in this order, mechanistic details about major effectors are explained after the experiments to probe these mechanistic details. For instance, with furin, this cleaves the spike protein to prime entry, and this is necessary information to understand why cells producing the pseudovirus were pre-treated with miR-24 in fig 6. While we agree the information is appropriate for the discussion, we believe including this background information in the introduction improves the readability and understanding of the specific experimental workflow.
Comment 4. In line 125, please clarify which lineage this SARS-CoV-2 belongs to, the original Wuhan or delta / lambda or omicron ?
Response 4: The SARS-CoV-2 isolate used for infections belongs to the original Wuhan strain. We have added this important detail to the methods section of our paper (line 126).
Comment 5. Line 137, the manufacturer should be Agilent instead of Aligent. Were mutations confirmed by sequencing ?
Response 5: Mutations were confirmed by Sanger sequencing at Génome Québec, Montréal. We have added this detail to our manuscript (Line 142-144).
Comment 6. In Line 155, did the authors do dose response studies. Are they transfecting miR-24-5p or miR-24-3p ? This should be made crystal clear throughout the manuscript.
Response 6: We did not perform dose response studies. However, our previous work with microRNAs in antiviral contexts has shown that 100 nM is sufficient to induce changes in expression of downstream targets (as demonstrated in Figure 3 and 6). The minimum miRNA mimic concentration required per the manufacturer is 0.3 nM suggesting that lower concentrations may also be effective, however, we have found in our experience that 50 nM or 100 nM tend to be very effective. (See Filip et al., 2021; Desrochers et al., 2022; Ahmed et al., 2023 as recent examples from our group).
Throughout the study, we used miR-24-3p. We have edited the manuscript to make this clearer.
Comment 7. Line 165 A549ACE2 should be A549-ACE2 cells. Please refer where these cells were obtained from or if they were generated in the lab and how.
Response 7: The A549 cells were originally obtained from ATCC and the A549-ACE2 line was generated from these using a stable expression lentiviral plasmid (Addgene #155295) and expression was confirmed via western blot. The methods have been updated to reflect this:
“A549 cells stably expressing ACE2 (A549-ACE2) cells were generated using a lentiviral, pLENTI_hACE2_PURO plasmid (Addgene #155295), a gift from prof. Raffaele De Francesco. Puromycin was used for selection and stable expression was confirmed via western blotting.”
Comment 8. Were miR-24 expression levels determined in untransfected cells prior to infection ? And following infection ? Is miR-24 deregulated by SARS-CoV-2 infection ?
Response 8: miR-24 levels were not measured in untransfected cells before or after infection. We do know that transfection with 100 nM miR-24 was sufficient to attenuate infection but the exact final concentration in the cells following the transfection is not known.
Consideration on whether miR-24 is deregulated by SARS-CoV-2 is an interesting point. Browsing the literature, there is limited published research on the effect of SARS-CoV-2 on intracellular miR-24; however, Gambardella et al., 2021 (Ref. 28) determined that levels of miR-24 in secreted extracellular vesicles were decreased in patients with COVID-19. The mechanism behind this was not explored in detail, however, it does appear that SARS-CoV-2 seems to attenuate miR-24 in some capacity.
Comment 9. Add Calu3 to Line 165 as well; else that reference appears out of place while reading the manuscript.
Response 9: A forward transfection was used for the faster growing cell lines (Huh7s and A549s) while a reverse transfection (with more cells) was used for the slower growing cell lines to ensure similar confluency. Huh7s and A549s were transfected the day after seeding at a relatively low density (30,000 cells per well). Conversely, Calu-3 cells were reverse transfected at a relatively higher density (80,000 cells per well). The text has been modified as follows to improve readability:
“To study the effect of a miRNA on the entry of pseudovirions, Huh7 cells and A549-ACE2 cells were seeded in 24-well plates and transfected the following day with 100 nM of miR-24 mimic, or a negative control miRNA mimic. Alternatively, Calu-3 cells were reverse transfected in 24-well plates with 100 nM of miR-24 mimic or a negative control, control miRNA mimic.”
Comment 10. Line 181, declare what DC assay is ? Was it a denaturing or native PAGE ?
Response 10: DC assay is a Detergent Compatible assay, which we used to quantify proteins before loading 40µg of lysates into 10% TGX stain-free gels, for denaturing sodium dodecyl sulfate polyacrylamide gel electrophoresis (SDS-PAGE).
We have updated our manuscript to make this more clear:
“Following protein quantification by detergent-compatible (DC) assay (Bio-Rad), 40 µg of lysates were loaded into 10% TGX stain-free gels (Bio-Rad) for denaturing sodium do-decyl sulfate (SDS) polyacrylamide gel electrophoresis (PAGE). Migrated proteins were transferred onto a PVDF membranes using the Trans-Blot turbo (Bio-Rad).”
Comment 11. Please include details of primary and secondary antibodies (isotype, manufacturer and dilution used) in the methods to allow reproducibility.
Response 11: The manufacturer, catalogue # and dilution used for each antibody have now been added under the “Detection of proteins by western blotting” section:
“The following antibodies and dilutions were used: The antibodies and manufacturers used are as follows: NRP2 (Thermo Fisher; PA5-75451) 1:1000, Furin (Thermo Fisher; PA5-96680) 1:1000, SREBP-2 (Thermo Fisher; PA5-88943) 1:2500, SARS-CoV-2 spike S10 (1A9) (GeneTex; GTX632604) 1:1000. Membranes were then incubated with Jackson ImmunoResearch secondary donkey-anti-rabbit antibody conjugated with horseradish peroxidase (1115-035-152), or goat-anti-mouse antibody conjugated with horseradish peroxidase (1115-035-062) depending on the identity of the primary antibody, at a 1:20000 dilution.”
Comment 12. Line 200. Have the authors verified that GAPDH levels are unaffected by transfection / SARS-CoV-2 infection ?
Response 12: GAPDH values remained very stable between samples during miR-24 transfection as evidenced by the RT-qPCR results for HCoV-229E. In section 3.7, we have modified text and included a Supplementary Figure 4 to demonstrate the stability of GAPDH during treatment with either miR-24 + HCoV-229E, con-miR treatment + HCoV-229E, no treatment + HCoV-229E, or no treatment + no infection.
GAPDH values were not measured during SARS-CoV-2 infection as the TCID50 method using a standard curve of diluted SARS-CoV-2 stock was used to normalize the data (Paquette et al., 2015 (Ref. 54) & Francis et al., 2021 (https://doi.org/10.1371/journal.ppat.1009705)).
Comment 13. Line 204, declare viral RNA before using vRNA. Please state cycling parameters to ensure reproducibility.
Response 13: We have modified the manuscript to make this more clear.
Comment 14. Line 209, were viral genome equivalents determined by quantitative PCR with standard curves. If yes, please include statement that MIQE guidelines were adhered to.
Response 14: Line 209 describes the RT-qPCR for SARS-CoV-2, the standard procedures and primers RT-qPCR clinical testing were employed as stated in Corman et al., 2020 (Ref. 55). The authors on this paper performed a standard curve with diluted SARS-CoV-2 to normalize the data as did we in the current paper. We have, therefore, added that MIQE guidelines were adhered to for this data.
Comment 15. Line 213, please include a statement on how miRDB assigns scores to predicted targets. In line 215, include a rationale for choosing 50 as a cutoff.
Response 15: The rational has now been included in the text. Briefly, the creators of the miRDB bioinformatic tool, Chen & Wang, 2019 (Ref. 29), define a score of 50 as the cutoff for a gene to be considered a predicted target.
Comment 16. Line 220, unaired should be unpaired. In my opinion, the correct statistical method here would be a one-way/ two-way ANOVA with post-hoc Geisser correction. Please state version of GraphPad Prism used.
Response 16: We fixed the typo, and added the version of GraphPad Prism used (10.1.1).
Regarding the statistical method, in the current work, the statistical tests are only performed between two unrelated groups. For this reason, we believe an unpaired t-test is more appropriate than a one-way/two-way ANOVA.
Comment 17. Results section 3.1 only describes what was done but does not discuss the data that were obtained in Figure 1. The authors should briefly state the objective of the section and then elaborate on the results obtained.
Response 17: The experimental details were provided following the description. The details of this section have been modified to improve the clarity and add more interpretation of the results:
“Following infection, supernatant was collected separately to assess produced virions (Figure 1B) while the cells were collected and lysed to probe for viral replication (Figure 1C). Compared to the control scramble miRNA (con-miR), miR-24 led to significant decreases in both viral replication (intracellular vRNA) and released virions (supernatant vRNA) at all time points assayed between 24-72h. Interestingly, the decrease in supernantant, or extracellular vRNA, was more pronounced than intracellular vRNA; this may be due to the lag time between replication and viral assembly and release or it may indicate additional inhibition at virion release. We also confirmed that miR-24 and con-miR were not cytotoxic by MTT assay (Supplementary Figure 1). The findings suggest that miR-24 is antiviral against SARS-CoV-2 by inhibiting at least one, but potentially more, stages of viral infection given that both intracellular and extracellular vRNA was reduced.”
Comment 18. Figure 1B Legend miR24 should be miR-24-**(the 5p or 3p miRNA used).
Response 18: We have made this clearer in the manuscript.
Comment 19. Line 243, did the authors check if miR-24 can directly target the SARS-CoV-2 genome ?
Response 19: Previous work done by our lab (Hum et al. 2021). shows that miR-24 has a seed binding site in the coding sequence for Spike of SARS-CoV-2 (Line 383). MicroRNAs that bind the coding sequence of mRNAs are typically less effective at inhibiting the expression of target genes. This corresponds well with our data, where there is not a decrease in expression of Spike, but there is a decrease in expression of the genes that contain a seed binding site for miR-24 in their 3’ UTRs, such as Furin, NRP2 and SREBP2 (Figure 6).
Comment 20. In Figure 3, please show the mRNA UTR on top and the miRNA below it. Show Watson-Crick pairs with straight lines and non-WC pairs with colon symbols. Indicate nucleotide positions in the 3'-UTR that are targeted. Make a graphic / table showing all mRNA: miRNA alignments ; the authors can sort them in descending stability based on free energy calculations.
Response 20: Figure 3 and the caption have been modified by removing the miRNA-mRNA graphic, instead moving it to supplemental. In addition, as requested by the reviewer, we have included mRNA:miRNA alignments for each gene, which show where miR-24 binds in the 3’UTR of each gene. These alignments show Watson-Crick pairs with straight lines in the seed site and those outside the seed site with colon symbols. We have taken it a step further and instead used the predicted binding interactions from miRDB for the seed sites. For interactions outside the seed, we have used Oligo Analyzer. In addition, per the reviewer’s comment, we have also added ∆G for each interaction, calculated by Oligo Analyzer. Some genes such as SREBP2 have several seed sites in different regions of the mRNA. Given that 3’UTR seeds are the most physiologically relevant, we have included only these sites and have chosen at least one site for each gene. This data is represented in Supplementary Table S3.
Comment 21. For controls, please show error bars.
Response 21: Regarding error bars on controls, the luciferase data in Fig. 6 was originally normalized to the control causing the error for both to be represented only on the miR-24 condition. The data was reanalyzed taking the average reading and the error bars have been added.
Comment 21b. Please replace miR-24 with miR-24-3p since that seems to be the miRNA studied in the manuscript.
Response 21b: We appreciate the distinction as this improves reproducibility. Accordingly, we have modified it in-text per the previous comment regarding this distinction.
Comment 21c. Have the authors considered the effect of miR-24-3p pretreatment on the cell cycle? Since miRNA transfections raise endogenous miRNA levels to non-physiological levels, please check for cytotoxicity and include that as supplementary data. miR-24-3p is a known regulator of cell cycle and this may alter viral replication. For many viruses, a particular stage of the cell cycle is preferred for infection. The reduction in viral titers could be due to alteration in cell cycle as an additional mechanism.
Response 21c: Regarding the non-endogenous levels of miR-24 and toxicity, per the reviewer’s suggestion, we have included a supplemental figure demonstrating the lack of toxicity during miR-24 transfection via MTT assay and have mentioned it in-text in section 3.1.
While miR-24-3p is a regulator of the cell cycle and this has been studied in papers such as that by Lal et al., 2009 (10.1016/j.molcel.2009.08.020) on K562 cells, these findings were not reflected in the MTT assay our group has performed. A possible interpretation of this result is that Calu-3s may not be as susceptible to miR-24-3p effects on cell cycle as other cell lines may be; however, it should also be noted that the MTT assay was used as a gauge on toxicity and we have not explored the effects of miR-24-3p on Calu-3 cell growth and how this may affect SARS-CoV-2. It is possible that other factors may contribute to the miR-24-mediated attenuation of SARS-CoV-2 as miRNAs can target many different genes.
Comment 21d. The statement in Line 315-316 is only partly substantiated by the data.
Response 21d: Regarding line 315-316 being only partially substantiated by data, which reads “These findings indicate that miR-24 inhibits the entry of SARS-CoV-2”. We believe that the claims are reasonably substantiated. We observe a roughly 75% decrease in entry of pseudovirus in Calu-3 cells and close to 90% decrease in entry of pseudovirus in Huh7s cells. SARS-CoV-2 pseudovirions provide information on entry of the virus through expression of luciferase in entered cells, therefore, a large reduction in luciferase activity indicates a large reduction in entry. It is therefore, reasonable to state that miR-24 is inhibitory towards SARS-CoV-2 entry.
Comment 21e. Have the authors considered infecting the cells and then transfecting with mimic transfection ?
Response 21e: Regarding transfection following infection, given the shorter timescales on in vitro studies with cells we chose to transfect beforehand to ensure that the miRNA has adequate time to downregulate targets to impede infection. We agree that both transfection after or before infection are suitable and useful metrics, evidenced by the application of both in previous miRNA studies. We chose to transfect before infection to 1) ensure proteins with longer turnover rates could also be studied in the viral context and 2) allow for an unbiased scope with maximal inhibition. This transfection order/sequence employed ended up being very useful for focused study on viral entry, given that these targets could downregulated prior to addition of virus or pseudovirus.
We agree that future work with expanded scope and work looking to establish the therapeutic efficacy for in vivo would benefit from infection followed by transfection.
Comment 21f. Or treat cells with anti-ACE2 or anti-SARS2 antisera in miR-24 transfected cells ? If entry is the key step regulated by miR-24, the authors would see no difference when cells are infected prior to transfection.
Response 21f: The reviewer raises an interesting point. Antisera can serve a useful role in studying entry. While we identify entry as the major step being identified given the downregulation of key coreceptors, we also allude to the possibility of regulation at other steps.
Antisera added prior to infection would inhibit the subsequent infection steps, serving a role as a negative control similar to our no infection controls we routinely perform during the infection but did not include in the datasets.
Alternatively, it could be possible to add it shortly after infection to then gauge intracellular replication. We do show this to some degree in Figure 1B for SARS-CoV-2 when assessing intracellular RNA although there still are cells being infected during this process. Our observations in subsequent experiments, however, with pseudovirions allow for isolated study of the entry (as they are non-replicative). Therefore, while we certainly agree on the utility of this suggestion to use antisera, our pseudovirus and non-infection controls present findings consistent with entry being a key step regulated by miR-24.
Comment 21 g. Additionally, ACE2 is a key receptor for SARS-CoV-2 along with TMPRSS2. Does miR-24 mimic transfection prior to infection alter ACE2 / TMPRSS2 transcript / protein expression?
Response 21g: This is an important question given that ACE2 and TMPRSS2 are inhibitory targets which have been well-studied in the context of SARS-CoV-2 in addition to the factors we have studied in our paper. We chose, however, to focus on well-studied targets which are also predicted direct targets of miR-24 and those that have been explored as targets experimentally. ACE2 and TMPRSS2 are not predicted targets of miR-24 and thus, we chose to focus our efforts on NRP1/2, furin and SREBP2 which were predicted targets and/or contained miR-24 seed sites.
Comment 21h. Did the authors validate overexpression of miR-24 in transfected cells by qPCR relative to a housekeeping gene ?
Response 21h: We did not determine the expression of miR-24 following transfection, however, the same final concentration of miRNA was transfected in each experiment and it was sufficient to cause inhibition and downregulation. Estimating intracellular miRNA concentrations can be somewhat error prone because we have previously observed transfection complexes trapped on the plasma membrane through cell imaging (unpublished data). While not entering the cell or causing an effect, these RNA complexes are still extracted with the cells potentially overestimating intracellular miRNA concentration. Given the difficulties with this, we believe that the total RNA transfected is a more appropriate measure. We do agree, however, that information on intracellular miRNA is useful and suggest that this will be certainly beneficial for future, in vivo work.
Comment 22. Did the authors titer the pseudovirus in Figure 4 and what MOI infections were done in Figure 4?
Response 22: We did not titer the pseudovirus in Figure 4. However, we confirmed that pseudovirus was being generated following the transfections outlined in Figure 4A by running supernatants on western blot, and probing for SARS-CoV-2 Spike and p24 (Figure 4B). Following the procedure outlined in Figure 4A, Huh7 cells or Calu-3 cells were infected with 100µL of supernatant containing pseudoviruses (Figure 4C and D).
Comment 23. Line 342, there is a typo ACE2 is spelled as AEC2.
Response 23: We have fixed this typo in the manuscript.
Comment 24. Data from Figure 5A does not demonstrate increased fitness of the D614G or the N501Y mutants compared to wild type virus in Huh7 cells. Whereas the wild type virus replicates well and is much more significantly inhibited in Huh7 cells, the degree of inhibition for the mutant viruses is at best modest. The effect is certainly more pronounced in A549 cells stably expressing human ACE2. The authors should also compare the level of miR-24 inhibition between D614G and the N501Y relative to controls using one-way ANOVA.
Response 24: We thank and agree with the reviewer that the data does not demonstrate increased fitness. We have removed the sentence suggesting D614G may have increased inhibition.
Experiments with D614G (and WT) and N501Y (and WT) were performed on separate days with biologically independent samples and, therefore, we do not feel comfortable performing statistical analysis by one-way ANOVA to compare them. Upon inspection, the values do appear somewhat similar and it is likely that any difference is non-significant or biologically irrelevant if the samples were to be compared.
Comment 25. Please use either con-miR or Con-mir consistently throughout the manuscript. Did the authors titrate the pseudovirus supernatants before infection ? What was the effective MOI or TCID50 for these experiments ?
Response 25: The supernatant containing pseudovirus was not titred before infection. Instead, equal volumes of supernatants (produced from equal seeding densities) were used to infect Huh-7, Calu-3, or ACE2-A549 cells.
In cases where the entry of pseudoviruses was being measured in Huh7 cells treated with con-miR as compared to miR-24, all cells were infected with equal volumes of supernatant containing pseudovirus that was generated from the same plate of pseudovirus-producing HEK293T cells (Figure 4 and 5).
In cases where the differential release and infectivity of pseudoviruses generated by con-miR or miR-24 treated HEK293T cells was being assessed, supernatants containing pseudovirus were collected and equal volumes were used to infect Huh7 cells (Figure 6).
Comment 26. Line 454, antimere is a typo of antimer
Response 26: We have fixed this typo in the manuscript.
Reviewer 2 Report
Comments and Suggestions for Authors
Evers et al. sought to elucidate the potential antiviral role of hsa-miR-24-3p in the context of SARS-CoV-2 infection. They first demonstrated that in vitro transfection of miR-24 in human cell lines results in decreased SARS-CoV-2 replication and virion production. Using in silico predictive tools, the authors identified several putative high-confidence host mRNA targets of miR-24 that have previously been shown to have proviral roles (e.g. as entry factors) for several virus infection models, including for SARS-CoV-2; namely, NRP-1, NRP-2, furin, and SREBP2. The authors systematically assessed the effect of exogenous miR-24 suppression of these host targets on SARS-CoV-2 infection. Importantly, the antiviral effect of miR-24 was preserved when assayed against two common SARS-CoV-2 mutants. Conversely, replication of HCoV-229E, a human endogenous coronavirus, was not negatively impacted by miR-24, owing to its independence from the previously mentioned host-factors. This study serves as an important proof-of-concept for miR-24 as a potential therapeutic for SARS-CoV-2 and other viral infections. The manuscript is extremely well written. I have only a few extremely minor suggestions below.
Line 220: Should be “unpaired”
Line 224 and others: Forgive me for being pedantic. The use of “in cellulo” to denote in vitro assays performed in cell/tissue culture systems appears to be a recent trend. This reviewer would suggest using the more familiar (and still considered accurate) term “in vitro”, but I will leave this to the authors to decide. For what it’s worth, it doesn’t even appear that “in cellulo” fits with Latinized nomenclature (“cellula” has been suggested as the correct term).
Line 371: Did you mean “…in virus producing cells…”?
Line 380: Change to “The S mRNA does have a predicted miR-24 seed binding site…”
Author Response
Reviewer #2:
Comments and Suggestions for Authors
Evers et al. sought to elucidate the potential antiviral role of hsa-miR-24-3p in the context of SARS-CoV-2 infection. They first demonstrated that in vitro transfection of miR-24 in human cell lines results in decreased SARS-CoV-2 replication and virion production. Using in silico predictive tools, the authors identified several putative high-confidence host mRNA targets of miR-24 that have previously been shown to have proviral roles (e.g. as entry factors) for several virus infection models, including for SARS-CoV-2; namely, NRP-1, NRP-2, furin, and SREBP2. The authors systematically assessed the effect of exogenous miR-24 suppression of these host targets on SARS-CoV-2 infection. Importantly, the antiviral effect of miR-24 was preserved when assayed against two common SARS-CoV-2 mutants. Conversely, replication of HCoV-229E, a human endogenous coronavirus, was not negatively impacted by miR-24, owing to its independence from the previously mentioned host-factors. This study serves as an important proof-of-concept for miR-24 as a potential therapeutic for SARS-CoV-2 and other viral infections. The manuscript is extremely well written. I have only a few extremely minor suggestions below.
Comment 1. Line 220: Should be “unpaired”
Response 1: We have fixed this in the manuscript.
Comment 2. Line 224 and others: Forgive me for being pedantic. The use of “in cellulo” to denote in vitro assays performed in cell/tissue culture systems appears to be a recent trend. This reviewer would suggest using the more familiar (and still considered accurate) term “in vitro”, but I will leave this to the authors to decide. For what it’s worth, it doesn’t even appear that “in cellulo” fits with Latinized nomenclature (“cellula” has been suggested as the correct term).
Response 2: We appreciate this suggestion. To ensure the article is more broadly accessible to readers, we have changed all instances of in cellulo to in vitro.
Comment 3. Line 371: Did you mean “…in virus producing cells…”?
Response 3: We have fixed this in the manuscript.
Comment 4. Line 380: Change to “The S mRNA does have a predicted miR-24 seed binding site…”
Response 4: We have fixed this in the manuscript.
Reviewer 3 Report
Comments and Suggestions for Authors
The authors studied the implication of miRNA miR-24-3p against SARS-CoV-2, through inhibition of viral entry, replication and production. The authors identified the host targets of miR-24 in the context of viral inhibition and that this inhibition is conserved against some of the common escape mutants. The manuscript is nicely written with sufficient experimental designs to test the hypothesis. However, there are several concerns/recommendations needed to be addressed:
1. Not sure if the word "proviral" used throughout the text is appropriate, might be confusing for people in HIV research as the word generally means integrated viral DNA in host chromosomes.
2. Line 287, the authors claimed that the NRP1, NRP2, Turin and SREBP2 as the "most promising" effectors which they later sought to validate experimentally, how did they narrow down to these four? Was there a p-value/ magnitude calculations from the bioinformatics approach to confirm? In the discussion other factors were mentioned, though they did not explore like ACE2 and SR-B1.
3. Fig 5, was the relative luciferase value calculated by normalization to the original strain? It was not mentioned in figure legend.
4. Fig 6B, except NRP2 which showed obvious reduced expression, others I can't tell visually (unlike what was stated in line 405), any quantification of protein expression done (using Image J etc)? The authors said S2' was reduced, along side other host proteins, how many fold reduction and how many biological replicates were performed?
5. Fig 6C, would include p24 western blot.
6. Fig 6D, how was the pseudovirus being quantified here? The figure legend mentioned S-pseudovirus entry assay was performed which is confusing.
7. Lastly, the authors mentioned the potential of a multitarget strategies with this miRNA-based antiviral approach. Since the targets of this miRNA are host factors, the authors might want to discuss the physiological roles of these proteins and any potential side effects if they were downregulated.
Author Response
Reviewer #3
Comments and Suggestions for Authors
The authors studied the implication of miRNA miR-24-3p against SARS-CoV-2, through inhibition of viral entry, replication and production. The authors identified the host targets of miR-24 in the context of viral inhibition and that this inhibition is conserved against some of the common escape mutants. The manuscript is nicely written with sufficient experimental designs to test the hypothesis. However, there are several concerns/recommendations needed to be addressed:
Comment 1. Not sure if the word "proviral" used throughout the text is appropriate, might be confusing for people in HIV research as the word generally means integrated viral DNA in host chromosomes.
Response 1: We appreciate the distinction and suggestion. To improve clarity, we have replaced instances of “proviral” to “promote viral infection” or “promote infection” where appropriate.
Comment 2. Line 287, the authors claimed that the NRP1, NRP2, Turin and SREBP2 as the "most promising" effectors which they later sought to validate experimentally, how did they narrow down to these four? Was there a p-value/ magnitude calculations from the bioinformatics approach to confirm? In the discussion other factors were mentioned, though they did not explore like ACE2 and SR-B1.
Response 2: The wording of “most promising” and the comments regarding it are very valid and we believe the language can be ameliorated. We have modified the language in text to read:
“We next sought to validate that the entry-related effectors, NRP1, NRP2, furin, and SREBP2, which have been mechanistically well-studied in the context of SARS-CoV-2 [16,30,34-36,38-41,46].”
Targets identified using bioinformatics for miR-24 were then cross-referenced with literature surrounding viruses, and specifically SARS-CoV-2, and then were chosen through a hypothesis-driven approach. As we explored deeper, we began to see a trend with targets related to cell entry while also being targets of miR-24 more directly. Given the greater abundance of mechanistic experimental work on these factors, we chose to explore them in greater detail. ACE2 was not a predicted direct target so this was not explored in detail. The reviewer, however, does note a very good point about SR-B1 being mentioned but not studied directly. SR-B1 has been studied in an excellent paper by Wei et al., 2020 (Ref. 64) and has also previously been studied as a miR-24 target in Ren et al., (https://doi.org/10.1016/j.atherosclerosis.2018.01.045) therefore, we focused our attention on other effectors.
Comment 3. Fig 5, was the relative luciferase value calculated by normalization to the original strain? It was not mentioned in figure legend.
Response 3: The relative luciferase data was calculated with the following equation dividing both the average con-miR and miR-24 values by the average con-miR value. The figure caption, as well as the others with luciferase data, has been updated to include this.
Comment 4. Fig 6B, except NRP2 which showed obvious reduced expression, others I can't tell visually (unlike what was stated in line 405), any quantification of protein expression done (using Image J etc)? The authors said S2' was reduced, along side other host proteins, how many fold reduction and how many biological replicates were performed?
Response 4: Western blots from 6B & 6C in the manuscript are representative of at least two biological replicates. To help quantify changes in protein expression of NRP2, Furin, and SREBP2, or changes in the abundance of S2’, we performed densitometry using ImageJ and included the densitometry as supplemental. This analysis shows the decrease in all studied proteins.
Comment 5. Fig 6C, would include p24 western blot.
Response 5: Unfortunately, p24 was not blotted for in this experiment. We blotted for p24 in the supernatant of Fig. 4 to show presence of pseudovirus in optimization our standardized procedure. After optimization steps, p24 was not blotted for in all pseudovirus supernatants. The reasoning for this is that “empty” pseudovirus can still be produced (as seen in Fig. 4) but this is not capable of entering host cells; however, blotting for SARS-CoV-2 S in supernatant demonstrates full pseudovirions given that free S is not secreted from the cell.
Comment 6. Fig 6D, how was the pseudovirus being quantified here? The figure legend mentioned S-pseudovirus entry assay was performed which is confusing.
Response 6: To improve clarity we have reworded the figure caption explaining the workflow. “HEK293T cells were pre-treated with either miR-24 or con-miR for 24h before transfection with the plasmids necessary for pseudovirus production. After 48h, the produced pseudovirus was collected from the supernatant of these HEK293T cells and filtred. To then quantify the amount of pseudovirus produced in either miR-24 or con-miR pre-treatment, the collected pseudovirus was then used to infect healthy, untreated Huh7s. The pseudovirus was then quantified via luciferase assay from untreated Huh7 cells.”
Comment 7. Lastly, the authors mentioned the potential of a multitarget strategies with this miRNA-based antiviral approach. Since the targets of this miRNA are host factors, the authors might want to discuss the physiological roles of these proteins and any potential side effects if they were downregulated.
Response 7: We agree that potential issues with targeting host factors are a very important observation and consideration when discussing antiviral therapies. To address these potential consequences, we have added a few sentences at the end of the discussion following discourse on the targets. We also do, however, acknowledge that an inhibitor of furin in phase 2b clinical trials has had a favourable safety profile, suggesting that they have potential to be targeted with correct and careful dosing. The added paragraph is as follows:
“While all of these factors are enticing targets for therapy and potential multi-target strategies, it is important to address limitations of this approach. We previously describe the benefits of targeting host factors as antiviral therapies [72], however, there is also the drawback that, unlike viral proteins, host proteins often serve physiological roles in normal homeostasis. For instance, NRP1/2 play roles in regulating vascular endothelial growth factor (VEGF) governing angiogenesis [43]; furin is important for processing transformation growth factor β1 (TGF-β1), controlling cell proliferation and differentiation [90]; and SREBP2 is a key modulator in canonical cholesterol homeostasis [49]. It should be noted, however, that there has been promising phase IIb safety profiles for an immunotherapy targeting furin (NCT02346747) [91], suggesting that with proper dosing, targeting these host factors could still be an effective therapy.”
Round 2
Reviewer 1 Report
Comments and Suggestions for Authors
The authors have addressed most of the concerns raised in the first draft successfully. The authors however do need to thoroughly proofread before submitting a revision. Below are comments on the revised version.
1. Line 144, please spell check. Agilent is now spelled as Aglient.
2. Comment # 6- The authors say that they have edited the manuscript and replaced miR-24 with miR-24-3p. This is incorrect. miR-24 (instead of the correct miR-24-3p) is used at multiple places beginning at line 19, 22, 24, 27, 68, 69, 176, 178, 246-248 and others. The reviewer would like to emphasize that the -5p and -3p isoforms can have completely different target repertoires and hence it is very important to be specific on which isoform is being tested and use that throughout the manuscript instead of making a few cursory changes.
3. Line 150-151. An address is not needed. End the sentence at Genome Quebec.
4. Line 198- duplication of partial sentence.
5. Line 229. Please do not end the sentence with a preposition. State that qPCRs were performed adhering to MIQE guidelines (add the reference).
6. Line 237. Please read the reference cited. The miRDB model uses machine learning based on CLIP-seq and RNAseq data to identify potential miRNA targets. The rationale for choosing a cut-off score of 50 is correct; the way it is worded needs to be rewritten.
7. Line 255- supernatant is spelled as supernantant. Please fix.
8. Line 274- State miR-24-3p in the legend heading instead of miR-24. The data show that it is miR-24-3p that was tested. The authors do not demonstrate the effect of miR-24-5p; hence the title should be specific to what was tested and what was observed.
9. Please address comment 20 thoroughly in the revision. Please indicate the nucleotide coordinates in the 3' UTR that are targeted by miR-24-3p. Please move the UTR above the miRNA (the norm is to depict a sequence from 5' to 3' end not otherwise) hence the UTR should be above not below. Please indicate the Watson:Crick pairing in the figure.
Comments on the Quality of English LanguageThe quality of English used is satisfactory.
Author Response
Comment 1. The authors have addressed most of the concerns raised in the first draft successfully. The authors however do need to thoroughly proofread before submitting a revision. Below are comments on the revised version.
- Line 144, please spell check. Agilent is now spelled as Aglient.
Response 1. The text has been fixed to read “Agilent”.
Comment 2. The authors say that they have edited the manuscript and replaced miR-24 with miR-24-3p. This is incorrect. miR-24 (instead of the correct miR-24-3p) is used at multiple places beginning at line 19, 22, 24, 27, 68, 69, 176, 178, 246-248 and others. The reviewer would like to emphasize that the -5p and -3p isoforms can have completely different target repertoires and hence it is very important to be specific on which isoform is being tested and use that throughout the manuscript instead of making a few cursory changes.
Response 2. We agree that it is important very important to be clear what isoform is being studied and we do acknowledge the great differences between variants. For reproducibility and clarity, we have replaced all instances of miR-24 in the entire article with miR-24-3p, including the title.
Comment 3. Line 150-151. An address is not needed. End the sentence at Genome Quebec.
Response 3. Done.
Comment 4. Line 198- duplication of partial sentence.
Response 4. We have edited the manuscript to read as follows:
“Membranes were then incubated with Jackson ImmunoResearch secondary donkey-anti-rabbit (1115-035-152) or goat anti-mouse (1115-035-062) antibody conjugated with horseradish peroxidase depending on the identity of the primary antibody, at a 1:20000 dilution”
Comment 5. Line 229. Please do not end the sentence with a preposition. State that qPCRs were performed adhering to MIQE guidelines (add the reference).
Response 5. We have fixed this in text and added in the original reference by Bustin et al., 2009 for Ref. 56.
The original text read as follows:
“The reactions were performed on a StepOnePlusTM Real-Time PCR System in a 96-well plate (Thermo Fisher) as previously described [54]. MIQE guidelines were adhered to.
The new text reads as follows:
“The reactions were performed on a StepOnePlusTM Real-Time PCR System in a 96-well plate (Thermo Fisher) as previously described [54], in accordance with MIQE guidelines [56].”
Comment 6. Line 237. Please read the reference cited. The miRDB model uses machine learning based on CLIP-seq and RNAseq data to identify potential miRNA targets. The rationale for choosing a cut-off score of 50 is correct; the way it is worded needs to be rewritten.
Response 6. We thank the reviewer for the comment and have added additional information to the methods. We believe this offers needed context into how miRDB scores mRNA targets. The modified paragraph is as follows:
"miRNA target prediction tool, miRDB [29], was used to determine predicted targets of miR-24-3p. The miRDB tool combines data from large-scale RNA-seq data (overexpressing 25 different miRNAs) along with direct miRNA:mRNA interactions determined through crosslinking immunoprecipitation (CLIP). The authors then trained a support vector machine (SVM) model, termed MirTarget, on these datasets to generate a prediction score, with those over 50 being considered predicted targets [29]. We conducted a search for miR-24-3p in miRDB, which yielded 959 targets scoring between 50-99. Targets with a score over 90 were chosen for the initial analysis. For subsequent analyses, targets were filtered by expression levels within each cell line studied as well as miRDB scores over 50, and those that were highly expressed (RPKM>20) were included in the Pathway Gene Ontology. Certain targets were taken from these lists in a hypothesis-based approach after cross-referencing with available literature."
Comment 7. Line 255- supernatant is spelled as supernantant. Please fix.
Response 7. Done. Fixed now in revised manuscript.
Comment 8. Line 274- State miR-24-3p in the legend heading instead of miR-24. The data show that it is miR-24-3p that was tested. The authors do not demonstrate the effect of miR-24-5p; hence the title should be specific to what was tested and what was observed.
Response 8. We have modified the legend to read miR-24-3p.
Comment 9. Please address comment 20 thoroughly in the revision. Please indicate the nucleotide coordinates in the 3' UTR that are targeted by miR-24-3p. Please move the UTR above the miRNA (the norm is to depict a sequence from 5' to 3' end not otherwise) hence the UTR should be above not below. Please indicate the Watson:Crick pairing in the figure.
Response 9. The nucleotide coordinates were included in the revised supplemental figure, however, we have added the word “nucleotide” and have moved 3’UTR to the beginning. As an example, NRP1 now reads: “3’-UTR nucleotide position 110-117”. In addition to this, we have added all of the predicted interactions between miRNA:mRNA in the 3-UTR and we have also used Targetscan (ref. 60) to indicate whether the site is conserved or not. miRDB does not include conservation in its prediction to ensure that predicted sites are included whether they are conserved or not conserved but this information may be useful to the reader.
In the revised SI, the UTR is currently above the miRNA in the figure and shown in the 5’ to 3’ direction. The miRNA is located below the the UTR. Watson:Crick pairing is also indicated in the figure by lines and colons represented non-Watson:Crick pairing;
“Watson-Crick pairs within the seed site are indicated by solid black lines (-) while Watson-Crick pairs and non-Watson-Crick pairs outside the seed site are indicated by a colon (:).”